**Data Availability Statement:** All relevant data that is essential to replicate the analyses and findings are presented in this paper and its Supporting information file.

# Investigation of the molecular biology underlying the pronounced high gene targeting frequency at the Myh9 gene locus in mouse embryonic stem cells

Lei Tan[1]☯, Yi Hu[1]☯, Yalan Li[1]☯, Lingchen Yang[1], Xiong Cai[2], Wei Liu[1], Jiayi He[1], Yingxin Wu[1], Tanbin Liu[1], Naidong Wang[3], Yi Yang[3], Robert S. Adelstein[4], Aibing Wang[1,4]*

**1** Laboratory of Animal Disease Prevention & Control and Animal Model, The Key Laboratory of Animal Vaccine & Protein Engineering, College of Veterinary Medicine, Hunan Agricultural University (HUNAU), Changsha, Hunan, China, **2** Institute of Innovation and Applied Research in Chinese Medicine, Hunan University of Chinese Medicine, Changsha, Hunan, China, **3** Laboratory of Functional Proteomics (LFP), The Key Laboratory of Animal Vaccine & Protein Engineering, College of Veterinary Medicine, HUNAU, Changsha, Hunan, China, **4** Laboratory of Molecular Cardiology (LMC), NHLBI/NIH, Bethesda, MD, United States of America

☯ These authors contributed equally to this work.
* bingaiwang@hunau.edu.cn

## Abstract

The generation of genetically modified mouse models derived from gene targeting (GT) in mouse embryonic stem (ES) cells (mESCs) has greatly advanced both basic and clinical research. Our previous finding that gene targeting at the Myh9 exon2 site in mESCs has a pronounced high homologous recombination (HR) efficiency (>90%) has facilitated the generation of a series of nonmuscle myosin II (NM II) related mouse models. Furthermore, the Myh9 gene locus has been well demonstrated to be a new safe harbor for site-specific insertion of other exogenous genes. In the current study, we intend to investigate the molecular biology underlying for this high HR efficiency from other aspects. Our results confirmed some previously characterized properties and revealed some unreported observations: 1) The comparison and analysis of the targeting events occurring at the Myh9 and several widely used loci for targeting transgenesis, including ColA1, HPRT, ROSA26, and the sequences utilized for generating these targeting constructs, indicated that a total length about 6 kb with approximate 50% GC-content of the 5' and 3' homologous arms, may facilitate a better performance in terms of GT efficiency. 2) Despite increasing the length of the homologous arms, shifting the targeting site from the Myh9 exon2, to intron2, or exon3 led to a gradually reduced GT frequency (91.7, 71.8 and 50.0%, respectively). This finding provides the first evidence that the HR frequency may also be associated with the targeting site even in the same locus. Meanwhile, the decreased trend of the GT efficiency at these targeting sites was consistent with the reduced percentage of simple sequence repeat (SSR) and short interspersed nuclear elements (SINEs) in the sequences for generating the targeting constructs, suggesting the potential effects of these DNA elements on GT efficiency; 3) Our series of targeting experiments and analyses with truncated 5' and 3' arms at the Myh9

**Funding:** This work was supported by General Program of National Natural Science Foundation of China (Grants Nos. 31571432 and 31802252) and Hunan Provincial Natural Science Foundation of China (Grant No. 2015JC3097). Support was also provided by "Shennong" Scholar funding to AW.

**Competing interests:** The authors have declared that no competing interests exist

exon2 site demonstrated that GT efficiency positively correlates with the total length of the homologous arms (R = 0.7256, p<0.01), confirmed that a 2:1 ratio of the length, a 50% GC-content and the higher amount of SINEs for the 5' and 3' arms may benefit for appreciable GT frequency. Though more investigations are required, the Myh9 gene locus appears to be an ideal location for identifying HR-related cis and trans factors, which in turn provide mechanistic insights and also facilitate the practical application of gene editing.

## Introduction

Transgenic animals that have had their genomes modified by genetic manipulation are extensively used to investigate the *in vivo* biological functions of genes as well as to mimic human diseases [1–5]. The mouse is the most widely engineered and commonly used organism for biomedical research. Its embryonic stem (ES) cell-mediated gene targeting has been a critical technology for generating such genetically modified mice [6, 7]. Moreover, the homologous recombination (HR)-based insertion, deletion or point mutation of the genome in ES cells and the consequent generation of targeted gain- and loss-of-function alleles have allowed the creation of thousands of mouse models for different purposes [1, 8, 9]. It is well known that the event of HR-mediated modification is rare. Even in mouse ES cells, obtaining the desired ES clone has therefore been a critical but time-consuming and labor-intensive step in previous studies [3, 10, 11]. Numerous factors including the length of the homologous arms, and the structure of the targeting vector, the targeted loci, the utilization of isogenic DNA and the status of ES cells, can affect the HR efficiency have been investigated and described [12–16]. However, the HR efficiency varies greatly even when the general principles are complied with. Furthermore, the conclusions from investigating the above factors may not be always consistent. For instance, the length of targeting arms is thought to be an important determinant of targeting efficiency [17], while increasing the targeting arm length does not always lead to the elevation of HR frequency [14]. These suggest the mechanisms underlying these influencing factors still remain elusive.

With the recent advent of a series of programmable nucleases or genome editing tools including zinc-finger nucleases (ZFN), transcription activator-like effector nucleases (TALEN) and Clustered Regularly Interspaced Short Palindromic Repeats (CRISPR)/CRISPR-associated protein9 (Cas9) (CRISPR/Cas9), the GT frequency in mammalian cells including mouse ES cells has been dramatically improved [18–23], suggesting the compatibility of artificial nucleases with traditional gene targeting methods. Therefore, these new technologies have stimulated scientific attention, since they have the potential of bypassing the requirement of using mouse ES cells in the generation of genetically engineered mice. However, these technologies are also accompanied by some disadvantages, in particular, the off-target effects and the inability to insert large DNA fragments [24], as further substantiated by a recent study [25]. In contrast, the time-consuming shortcoming of the traditional method has been overcome by a recent report, which demonstrated that the approach of gene targeting in mouse extended pluripotent stem (EPS) cells, coupled with tetraploid complementation technology, can generate mutant mice in approximately 2 months [26]. Additionally, targeted alterations of the genome in ES cells, created with or without the assistance of gene editing tools, are widely used for various purposes, such as the in vitro differentiation of ES cells and the screening of druggable chemicals [23]. These highlight the necessity to investigate the intrinsic properties of HR-based gene targeting in ES cells for further improving the efficiency and for providing valuable clues to apply gene targeting in other mammalian cells.

Our previous finding indicated that gene targeting at the exon2 site of the Myh9 gene which encodes the protein of nonmuscle myosin heavy chain IIA (NMHC IIA) has a pronounced high HR efficiency (>90%) in mouse ES cells. This unique property has been applied to the production of a series of nonmuscle myosin II (NM II) related mutant mouse lines (≥7) [27–30]. Moreover, this finding has also been extended to the site-specific insertion of other exogenous genes in mouse ES cells [31]. In the present study, we intended to investigate the molecular biology underlying this distinctive HR efficiency from multiple aspects.

## Materials and methods

This work was approved by the Animal Ethics Committee of Hunan Agricultural University, Hunan, China. No animal experiments were conducted, no ethic permits were therefore needed for this report, which complied with all of the relevant regulations.

### Bioinformatics analysis

A 3950 bp sequence immediately before and a 1327 bp sequence immediately after the Myh9 gene exon2 across various species (including mouse, human, chimpanzee, pig, rabbit, and rat) were retrieved from useast.ensembl.org and further confirmed with genome.ucsc.edu. The comparison and alignment of the sequence region (<6 kb) around the Myh9 gene exon2 across distinct species was performed by using Clustal W in the DNAStar version 7.10 software (Lasergene) [32]. The DNA features including GC-content, simple sequence repeat (SSR), short interspersed nuclear elements (SINEs), and long interspersed nuclear elements (LINEs), other DNA elements in the sequences used for creating targeting constructs were analyzed via Repeat Masker Web Server (http://www.repeatmasker.org/cgi-bin/WEBRepeatMasker). The CpG islands in these sequences were determined using CpG report software (https://www.ebi.ac.uk/Tools/seqstats/emboss_newcpgreport/).

### Generation of targeting constructs

The constructs targeting to the Myh9 gene exon2, intron2 and exon3 sites were separately generated by using the same strategy as described previously [27, 31]. Briefly, DNA fragments for the 5' and 3' homologous arms were amplified by PCR using the primer pairs listed in S1 Table and the template of 129/Sv genomic BAC clone DNA, and then cloned into the vector mpNTKV-LoxP described previously [27]. Likewise, the truncated 5' and 3' homologous arms were firstly amplified by PCR with the primer pairs listed in S1 Table and the template of 129/Sv genomic BAC clone DNA, and then various combinations of the 5' and 3' homologous arms were cloned into the mpNTKV-LoxP vector. Nucleotide sequences of the cloned DNA fragments were verified in all cases by sequencing. All targeting vectors were linearized by restriction enzymes before electroporation.

### Culture and electroporation of mouse ES cells

The mouse V6.5 ES cells used in this study were originally derived by Eggan et al [33] from F1 hybrid mice with 50% C57BL/6 and 50% 129/Sv mixed genetic background (Notably, the sequences of the targeting constructs are isogenic to both chromosomes from C57BL/6 and 129/Sv genetic background in terms of the Myh9 gene region involved.), cultured and electroporated with linearized targeting vectors as described before [27, 29], followed by drug selection using 400μg/ml G418 and 200μM ganciclovir. Drug resistant colonies were picked and expanded for the preparation of genomic DNA.

## PCR identification of recombination events

HR events occurring at the Myh9 gene exon2 (including those truncated targeting constructs), intron2 and exon3 sites were separately identified by PCR with the primer pair (the forward primer located at the neomycin, while the reverse primer resided immediately outside the 3' short arm) listed in S1 Table. The preparation of genomic DNA, the components of PCR system and the PCR reaction conditions completely followed the previous description [31]. The PCR products were analyzed by electrophoresis on 1.0% agarose gel, the wild-type allele produced no band while the targeted allele yielded an expected ≥2.1, 2.2, 2.3 kb band, respectively, as indicated in S1 Table. At least five randomly selected PCR-amplified products were excised, mixed, extracted and cloned into the T-easy vector for sequencing (Promega). The sequencing results could further confirm the gene targeting recombinants.

## Data analysis

The relationship between the length of homologous arms and the targeting efficiency, as well as the significance, were analyzed with GraphPad.Prism.v5.0 software (GraphPad Software inc, CA, USA).

# Results and discussion

## Bioinformatic analysis of the Myh9 gene locus and other widely used loci for site-specific integration of exogenous genes

As demonstrated in previous studies, gene targeting at the Myh9 gene locus, in particular the exon2 site, had a pronounced high HR efficiency (>90%) in mESCs [27, 29–31]. Although GT efficiency in mESCs is considered 10–100 fold higher than that in mammalian somatic cells [34]. We are not aware of any reports showing such a high GT efficiency in mESCs under similar experimental conditions. Thus, the Myh9 gene locus can be added to the list of a few identified mitotic HR hotspots [35–37]. This finding also evoked our further curiosity about the molecular biology underlying it. Furthermore, the mechanisms responsible for the higher relative rate of gene targeting in mESCs than those in somatic cells remain unclear. As an initial effort, we examined whether the Myh9 locus has some common genetic features with other widely used loci for site-specific integration of exogenous genes, including collagen alpha 1 (ColA1), hypoxanthine phosphoribosyltransferase (HPRT) and ROSA26. To this aim, we firstly collected and summarized targeting events occurring at these loci from the references, as indicated in Table 1. Though there were several differences in the length of 5' and 3' arms, the ratio of them, ES cell lines used, the selection markers utilized, appreciable targeting efficiencies were obtained in these loci. Without considering other factors, several characteristics could be observed: a) These loci have at least one unique property, such as only one copy of exogenous gene insertion at the HPRT locus, high expression of exogenous gene at the ColA1, Myh9 and ROSA26 loci, thereby being widely used for targeted transgenesis; b) Generally, a 5–6 kb total length of homologous arms was utilized, while longer total length of homologous arms did not mean higher targeting efficiency; c) It seemed that the length ratio of the 5' and 3' arms also matters, as further demonstrated in our subsequent experiments. Next, we examined whether there exist some common genetic features among these loci. For this purpose, the sequences used for generating the constructs targeting to these loci were analyzed as described in Materials and Methods, the analytical results based on the 5' or 3' arm were summarized in Table 2. As far as these sequences were concerned, several features could be generalized: 1) GC-content in these sequences for the 5' and 3' arms displayed a range of 38–68%, a similar and about 50% GC-content seemed better than other cases with regard to GT efficiency; 2)

**Table 1. Comparison of the HR efficiency and features of frequently targeted loci/sites.**

| Locus/Gene | Total length of homologous arms (kb) | Length of 5' arm (kb) | Length of 3' arm (kb) | Ratio of the length of 5' to 3' arm | ES cell lines | Positive/Negative selection marker | Targeting efficiency (%) | References |
|---|---|---|---|---|---|---|---|---|
| Myh9 | 5.7 | 4.0 | 1.7 | 2.3:1 | V6.5 | Neomycin/TK | 91.7 | [31] |
| ColA1 | 6.0 | 3.3 | 2.7 | 1.2:1 | V6.5 | Neomycin/- | 80.0 | [39, 40] |
| HPRT | 9.6 | 3.8 | 5.8 | 1:1.5 | BPES | HAT/- | **High** while detailed information undisclosed | [39, 41–44] |
| ROSA26 | 5.4 | 1.1 | 4.3 | 1:3.9 | AK7 | Neomycin/DTA | 34.8 | [39, 45] |
| | 5.4 | 1.1 | 4.3 | 1:3.9 | HM-1 | Neomycin/TK | Undisclosed | [46] |

TK: thymidine kinase gene; HAT: hypoxanthine-aminopterin-thymidine; DTA: diphtheria toxin A.

There existed different DNA elements in the 5' or 3' arm of these constructs, such as simple sequence repeat (SSR), short interspersed nucleotide elements (SINEs), long interspersed nucleotide elements (LINEs), even CpG island or other elements, the effects of these elements on targeting efficiency have also been suggested [38]; 3) Notably, different DNA elements were distributed in the 5' and 3' arms of them, no common genetic properties could therefore be identified from these sequences. Furthermore, the practical influences of these DNA elements on GT frequency remained to be substantiated.

## GT efficiency at various targeting sites of the Myh9 gene locus

The effects of loci, including chromosome position and sequence context on gene targeting efficiency are well recognized [17, 37], but the effects of shifting the position of homologous arms on the same locus remain to be explored. Undoubtedly, the property of high GT efficiency at the Myh9 locus is useful for investigating this aspect. For this purpose, a series of constructs targeting to various sites of the Myh9 gene locus including the exon2, intron2 and exon3 positions, were generated. Notably, the constructs targeting to the intron2 and exon3 sites even had slightly longer homologous arms as indicated in Fig 1A since the GT efficiency was suggested to be directly proportional to the length of the homologous arms. Gene targeting experiments with these constructs were performed and HR events were identified as described above (Fig 1B and 1C). In contrast to previous studies [16, 47, 48], our results demonstrated that longer arms are not always associated with higher GC efficiency and also revealed that there is obvious difference in the GT efficiency at various sites even in the same locus. Briefly, the highest GT efficiency occurred at the exon2 site (91.7%), followed by that at the intron2

**Table 2. Comparison of the sequence features of several widely used loci for site-specific insertion of exogenous gene.**

| Gene/locus | Length of the 5' and 3' homologous arms | G/C content (%) (5'/3') | SSR (%) (5'/3') | CpG islands (n) (5'/3') | Percentage of SINE (%) (5'/3') | Percentage of LINE (%) (5'/3') | Other DNA elements (%) (5'/3') |
|---|---|---|---|---|---|---|---|
| Myh9 | 4.0/1.7 | **50.4/50.8** | 0.0/0.0 | 0/0 | 3.1/0.0 | 0.0/0.0 | 0.0/0.0 |
| ColA1 | 3.3/2.7 | **51.7/49.7** | 5.9/2.9 | 0/0 | 0.0/0.0 | 0.0/0.0 | 0.0/0.0 |
| HPRT | 3.8/5.8 | 43.0/39.3 | 1.9/7.5 | 0/0 | 0.0/0.0 | 4.5/0.0 | 0.0/0.0 |
| ROSA26 | 1.1/4.3 | 68.3/38.7 | 0.0/0.0 | 1*/0 | 0.0/4.1 | 0.0/0.0 | 0.0/2.6 |

GC content represents the percentage of nucleotides in the strand that possesses either cytosine or guanine bases; Simple sequence repeat (SSR) consists of short, tandemly repeated di, tri-, tetra- or penta-nucleotide motifs; CpG island is a short stretch of DNA in which the frequency of the CG sequences is higher than other regions; SINEs denotes short interspersed nuclear elements; LINEs denotes long interspersed nuclear elements;

*CpG island located at the position of 48–810.

**A**

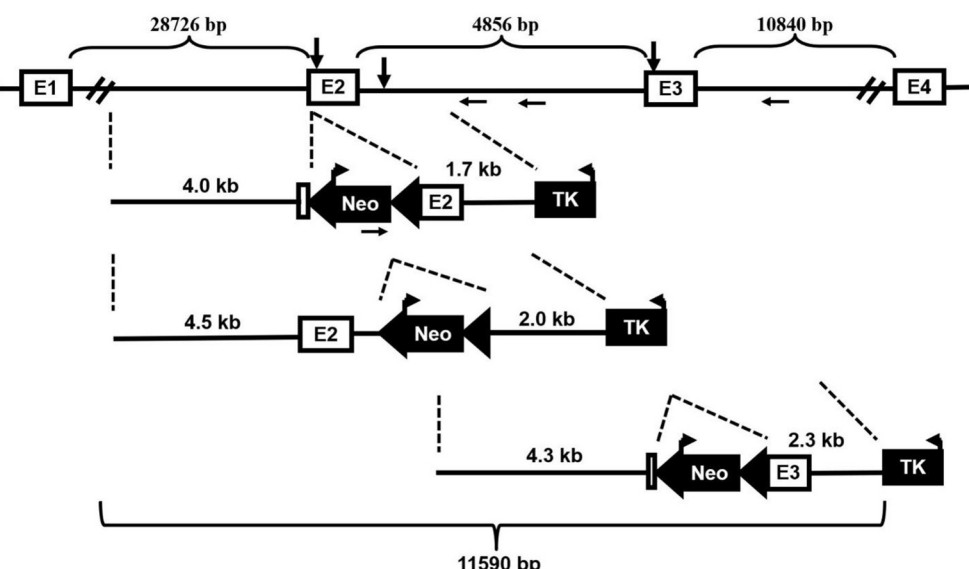

**B**

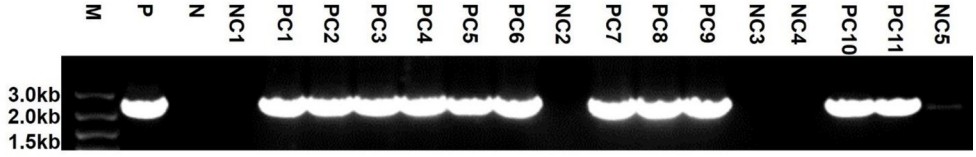

**C**

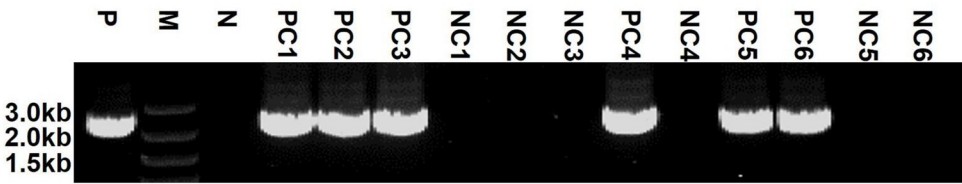

**Fig 1. Gene targeting at various sites of the mouse Myh9 locus. (A)** Diagram of the constructs targeting to different sites of the Myh9 gene locus. Shown are part of the Myh9 gene locus including exons1-4, the distance between them, the down arrows representing the targeted insertion of Neomycin (upper), the constructs targeting to distinct sites of the Myh9 gene locus and the length of the 5' and 3' arms of these constructs (middle), as well as the whole sequence region used for the generation of these constructs (lower). The HR efficiency of these targeting constructs is summarized in Table 3. The average percentage of HR efficiency is derived from two repeats. **(B)** Representative gel image of PCR identification of the GT recombinants at the Myh9 gene exon2 site. **(C)** Representative gel image of PCR identification of the GT recombinants at the Myh9 gene exon3 site. For representative gel image of PCR identification of the GT recombinants at the Myh9 intron2 site, please see reference [31]. M: DNA Marker; P: Positive control; N: Negative control; PC: positive clone; NC: negative clone.

site (71.8%), while the exon3 site had the lowest efficiency (50.0%), displaying a trend of gradual decrease with the shift of the targeting site from the exon2 to intron2 and exon3 (Table 3). Consistent with a previous idea that the nature of the locus is critical [17], our result further highlighted the importance of the suitable targeting site in the same gene locus for desirable GT frequency. Furthermore, this view was also supported by recent studies, which suggested that even the sites for the application of gene editing tools to some specific loci should be considered carefully [49, 50]. Notably, our previous studies indicated that in spite of high GT frequency at the Myh9 gene locus, both PCR and Southern Blot did not identify ES clones with double knockout alleles [27, 29–31]. The major reason might be that homozygous knockout of the Myh9 gene can cause abnormal proliferation and ESC morphology [51], which in turn result in bias for colony picking, so does the current study.

To explore the underlying reasons responsible for obvious difference in the GT efficiency at various sites of the Myh9 locus, the whole sequences used to create these targeting constructs were analyzed in detail as described in Materials and Methods. As summarized in Table 4, the GC-content in these three sequences for constructs targeting to the Myh9 exon2 (50.5%), intron2 (50.6%) and exon3 (51.1%) was only slightly increased with the extended length of the homologous arms, while the percentage of SSR and SINEs in them was markedly decreased. Briefly, the construct targeting to the Myh9 exon2 possessed the highest percentage of SSR (11.1%) and SINEs (9.2%), followed by those of the construct targeting to the intron2 (10.1%, 8.0%, respectively), while the construct targeting to the Myh9 exon3 had the lowest ones (4.2%, 2.3%, respectively). Interestingly, the trends of decreased percentage of SSR and SINEs in the homologous arms of these constructs were completely consistent with that of the GT efficiency for them. Importantly, the potential effects of these elements on gene targeting have been suggested [52–56], though direct evidence remain to be substantiated. Therefore, this finding presented here not only supports the concept, but also provide an excellent locus for in-depth investigating this issue in the future.

## GT efficiency at the Myh9 gene locus with various homologous arm lengths

Earlier investigations suggested that there is a good correlation between GT efficiency and the length of homologous arms [47, 56–59], but another report also showed that increasing the length of the targeting sequences does not always enhance the efficiency of HR [14]. This idea was partially supported by our observation that the construct targeting to the exon2 site of the Myh9 gene locus with a total length of approximate 5.7 kb homologous arms obtained an unprecedented HR frequency [27, 29, 30]. Additionally, our effort to use the constructs with increasing length of homologous arms targeting to distinct sites of the Myh9 gene locus also confirmed this concept, as demonstrated above. Next, two questions related to the targeting events at the Myh9 gene locus were raised about whether this phenomenon is associated with specific sequences in the homologous arms, how long the homologous arms and what the ratio

**Table 3. Gene targeting frequency at distinct targeting sites of the Myh9 locus in mouse ES cells.**

| Targeting site | Positive vs Screened clones | Targeting frequency (%) |
|:---:|:---:|:---:|
| exon2 | 10/12 | 91.7 |
| | 12/12 | |
| intron2 | 11/16 | 71.8 |
| | 12/16 | |
| exon3 | 6/12 | 50 |
| | 6/12 | |

**Table 4. The bioinformatic analysis of the sequences used to create the constructs targeting to different sites of the Myh9 locus.**

| The constructs targeting to the sites of the Myh9 locus | The properties of the sequences used to create the targeting construct | | | | | | |
|---|---|---|---|---|---|---|---|
| | Total length (bp) | GC content (%) | Percentage of simple sequence repeat (SSR) (%) | CpG Island (%) | Percentage of SINEs (%) | Percentage of LINEs (%) | Percentage of other DNA elements (%) |
| exon2 | 5630 | 50.5 | 11.1 | No | 9.2 | 0.0 | 0.0 |
| intron2 | 6485 | 50.6 | 10.1 | No | 8.0 | 0.0 | 0.0 |
| exon3 | 6580 | 51.4 | 4.2 | No | 2.3 | 0.0 | 0.0 |

for them are required for optimal GT efficiency. In particular, there is no exact answer for the latter one. On the one hand systematic studies have not been conducted, and on the other hand comparing results generated from different studies is problematic because of the presence of multiple variables in addition to the extent of homology [60]. To answer these questions, a series of truncated targeting constructs with various lengths of the 5' and 3' homologous arms were generated as described in Materials and Methods and indicated in Fig 2A. Similarly, PCR was utilized to identify targeting recombinants (Fig 2B). The GT frequency for this series of constructs was summarized in Table 5. Additionally, the trend and relationship of the length of homologous arms and the GT efficiency was analyzed as indicated in Fig 2C. The results indicated: a) With the shortening of the length of the homologous arms either at the 5' or 3', the GT frequency displayed a generally downward trend, suggesting the effect of the homologous arm length on the GT efficiency, as consistent with previous reports [57, 59, 60]. Furthermore, the total length of the homologous arms had a positive correlation with the GT frequency ($R^2$ = 0.7256, p<0.01); b) The shortening of 5' and 3' arms seemed to have different effects on the GT frequency, overall with that of the former more pronounced than that of the latter; c) Only when the truncated length of the 5' arm reached one fourth of the original length, the decrease of the targeting efficiency approached an order of magnitude. This differed from a previous report [47] and suggested the effect of the length of the homologous arms on the targeting frequency is complicated and may not be pronounced as anticipated [14]; d) The effect of the absence of the fragment containing the exon2 sequence which is more conserved than others in the 3' arm on the targeting efficiency was not as large as expected; e) A ratio of 2:1 for the length of the 5' and 3' arms seemed better than other combinations (Table 5). Collectively, a targeting construct with a 5.7 kb sequence surrounding the Myh9 gene exon2 as the homologous arms produces a pronounced high GT efficiency in mESCs, while this efficiency is correlated with the length of homologous arms but not to a specific sequence within them. Additionally, an optimal ratio of 2:1 for the length of the homologous arms is suggested even in the presence of gene editing tools for desirable targeting efficiency. Lastly, we examined whether the shortening of 5' or 3' homologous arm alters the amount of DNA elements in these sequences, which in turn influences GT efficiency. Similarly, the presence of different DNA elements in each truncated 5' or 3' arm was analyzed as above and summarized in Table 6. Firstly, the shortening of 5' or 3' arm seemed not to obviously change the GC-content. Secondly, only the shortening of 5' arm from 4 kb to 2 kb led to a markedly increase of the percentage of SINEs, which in turn alleviates the effect of the shortening of the 5' arm on GT efficiency. This fact was to some extent consistent with the idea from a previous report [38], which suggests the integration frequency of targeting vector correlates with the amount of SINEs present in the arms.

## Conclusion

In the current study we further investigated our previous finding that gene targeting at the Myh9 gene locus has a pronounced high HR frequency. Our study clarified several points and

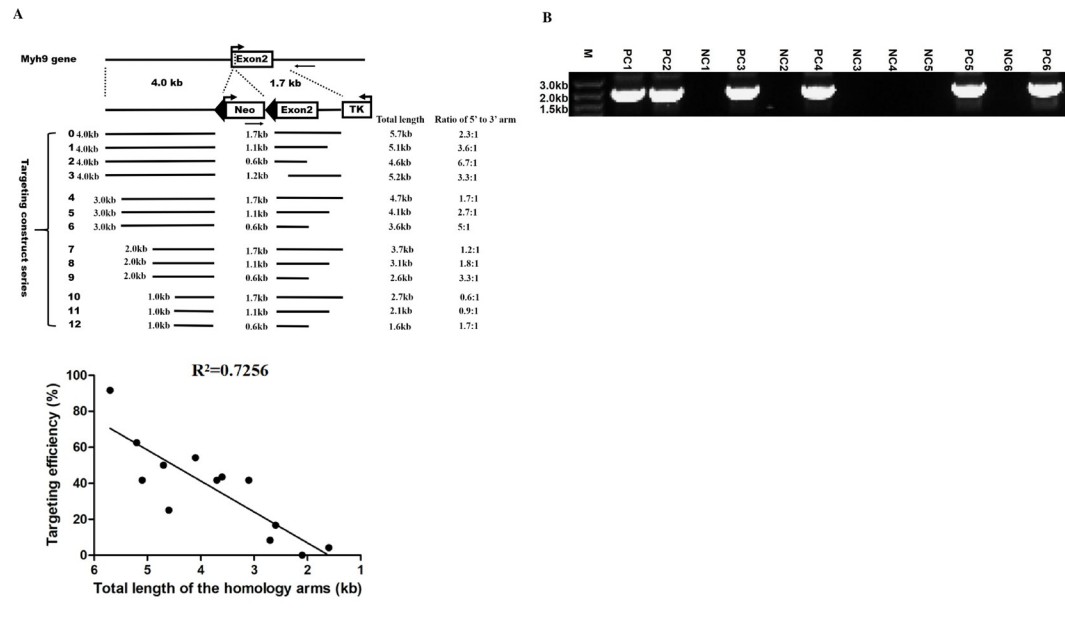

**Fig 2. Gene targeting at the Myh9 gene exon2 site with various length of the homologous arms. (A)** Diagram of a series of truncated targeting constructs. The 5' homologous arm is left-side truncated from 4, 3, 2 to 1 kb, while the 3' one is right-side truncated from 1.7, 1.1 to 0.6 kb. Additionally, a left-side truncated 3' arm with the deletion of the exon2 and beyond (about 0.5 kb) is also generated. Each targeting construct is given a number (No) as indicated at the left, the total length, the length of each 5' or 3' arm and the ratio between them are also indicated. Detailed information of these constructs and their HR frequency in mouse ES cells are summarized in Table 5. The average percentage of HR efficiency is derived from two repeats. **(B)** Shown is representative gel image of PCR identification of the GT recombinants for the targeting construct No4. M: DNA marker; PC: positive clone; NC: negative clone. **(C)** The graph indicates a downward trend of the gene targeting efficiency with the decreased length of the homologous arms and reflects the correlation of the gene targeting frequency with the total length of the homologous arms, the $R^2$ value shown in the graph is calculated from thirteen points (p<0.01).

provided some new insight into homologous recombination-based gene targeting. Firstly, through the analysis of the Myh9 and several other widely used loci for targeting transgenesis, the properties of a 6 kb total length, and 50% GC-content present in the homologous arms,

**Table 5. Summary of the HR frequency for each targeting construct.**

| Targeting Construct No* | Left arm length (kb) | Right arm length (kb) | Positive vs Total clones | Targeting frequency (%) |
|---|---|---|---|---|
| 0 | 4.0 | 1.7 | 22/24 | 91.7 |
| 1 | 4.0 | 1.1 | 10/24 | 41.7 |
| 2 | 4.0 | 0.6 | 6/24 | 25.0 |
| 3 | 4.0 | 1.2 | 15/24 | 62.5 |
| 4 | 3.0 | 1.7 | 12/24 | 50.0 |
| 5 | 3.0 | 1.1 | 13/24 | 54.2 |
| 6 | 3.0 | 0.6 | 10/23 | 43.5 |
| 7 | 2.0 | 1.7 | 10/24 | 41.7 |
| 8 | 2.0 | 1.1 | 10/24 | 41.7 |
| 9 | 2.0 | 0.6 | 4/24 | 16.7 |
| 10 | 1.0 | 1.7 | 2/24 | 8.3 |
| 11 | 1.0 | 1.1 | 0/24 | 0.0 |
| 12 | 1.0 | 0.6 | 1/24 | 4.2 |

**Table 6. Genetic features of the sequences used for creating different 5' and 3' truncated homologous arms.**

| Truncated arms (kb) | G+C content (%) | SSR | CpG island | Percentage of SINE (%) | Percentage of LINE (%) | Other DNA elements |
|---|---|---|---|---|---|---|
| **5'-4.0** | 50.4 | 0.0 | No | **3.1** | 0.0 | 0.0 |
| **5' 3.0** | 50.2 | 0.0 | No | **4.1** | 0.0 | 0.0 |
| **5'-2.0** | 51.0 | 0.0 | No | **6.1** | 0.0 | 0.0 |
| 5'-1.0 | 51.5 | 0.0 | No | 0.0 | 0.0 | 0.0 |
| 3'-1.7 | 50.9 | 0.0 | No | 0.0 | 0.0 | 0.0 |
| 3'-1.1 | 50.9 | 0.0 | No | 0.0 | 0.0 | 0.0 |
| 3'-0.6 | 54.5 | 0.0 | No | 0.0 | 0.0 | 0.0 |
| 3'-1.2 | 49.2 | 0.0 | No | 0.0 | 0.0 | 0.0 |

may facilitate the achievement of desirable GT efficiency. Secondly, consistent with the central role for the loci, our study also highlights the importance for the selection of an exact targeting site in the practical application of gene targeting, this may be associated with the amount of DNA elements like SINEs in the homologous arms. Thirdly, the targeting efficiency has a positive correlation with the total length of the homologous arms. Moreover, an optimal ratio of 2:1 for the length of the 5' and 3' arms with 50% GC-content and the potential effects of DNA elements including SINEs in the arms on GT frequency, are further suggested. Lastly, the in-depth investigations of gene targeting greatly benefit the understanding of the cellular machinery related to HR, the regulation of it, as well as HR-associated factors. The Myh9 gene locus as a potential mitotic recombination hotspot undoubtedly is an ideal molecular and genetical location for further exploration of these aspects.

## Supporting information

**S1 Table. The PCR primers used in this study.**
(DOCX)

## Acknowledgments

The authors acknowledge Dr. Wen Xie for the mES cell culture and electroporation and Dr Chengyu Liu and Mary Ann Conti for reading and revising the manuscript.

## Author Contributions

**Data curation:** Lei Tan, Yi Yang.

**Formal analysis:** Lei Tan, Yi Hu, Yalan Li, Lingchen Yang, Xiong Cai, Wei Liu, Jiayi He, Yingxin Wu, Tanbin Liu, Naidong Wang, Yi Yang.

**Funding acquisition:** Aibing Wang.

**Validation:** Robert S. Adelstein.

**Visualization:** Aibing Wang.

**Writing – original draft:** Lei Tan, Robert S. Adelstein, Aibing Wang.

**Writing – review & editing:** Robert S. Adelstein, Aibing Wang.

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
