## [Decision Letter · Decision Letter 0]

29 Aug 2019

PONE-D-19-15918

Investigation of the molecular biology underlying the pronounced high gene targeting frequency at Myh9 gene locus in mouse embryonic stem cells

PLOS ONE

Dear Dr Wang,

Thank you for submitting your manuscript to PLOS ONE. After careful consideration, we feel that it has merit but does not fully meet PLOS ONE’s publication criteria as it currently stands. Therefore, we invite you to submit a revised version of the manuscript that addresses the points raised during the review process.

We would appreciate receiving your revised manuscript by Oct 13 2019 11:59PM. To enhance the reproducibility of your results, we recommend that if applicable you deposit your laboratory protocols in protocols.io, where a protocol can be assigned its own identifier (DOI) such that it can be cited independently in the future. For instructions see: http://journals.plos.org/plosone/s/submission-guidelines#loc-laboratory-protocols

We look forward to receiving your revised manuscript.

Kind regards,

Wenhui Hu, M.D., Ph.D.

Academic Editor

PLOS ONE

Journal Requirements:

Reviewers' comments:

Reviewer's Responses to Questions

**Comments to the Author**

1. Is the manuscript technically sound, and do the data support the conclusions?

Reviewer #1: No

Reviewer #2: Partly

Reviewer #3: Partly

2. Has the statistical analysis been performed appropriately and rigorously? 

Reviewer #1: No

Reviewer #2: N/A

Reviewer #3: No

3. Have the authors made all data underlying the findings in their manuscript fully available?

Reviewer #1: No

Reviewer #2: No

Reviewer #3: No

4. Is the manuscript presented in an intelligible fashion and written in standard English?

Reviewer #1: Yes

Reviewer #2: Yes

Reviewer #3: No

5. Review Comments to the Author

Reviewer #1: In the manuscript entitled “Investigation of the molecular biology underlying the pronounced high gene targeting frequency at Myh9 gene locus in mouse embryonic stem cells”, the authors perform homologous recombination (HR) in mouse embryonic stem cells (ESCs), induced pluripotent stem cells (iPSCs), and embryonic fibroblasts (MEFs) to investigate the molecular features of the Myh9 gene locus that are responsible for the notably high efficiency of gene targeting. They perform comparative genetic sequence analysis of the Myh9 gene locus to illustrate a low degree of sequence conservation, which in turn suggests there are no unique genetic properties at these loci that affect recombination efficiency. By targeting their vector into the exon 2 region of Myh9 in mouse ESCs, iPSCs, and MEFs, the authors demonstrate a variable HR efficiency that is cell type-dependent but transcriptionally and translationally independent. The efficiency of HR also declines as the targeting vector is shifted downstream of exon 2, highlighting the contributions of local regions to recombination efficiency. Finally, the authors investigate the effects of modifying the lengths of the homology arms and discover an optimal ratio of 5’ to 3’ arm lengths for effective HR. Because gene editing has proven itself as an indispensable tool in biomedical research, primarily in enabling the generation of genetic models of development and disease, it is a worthwhile effort to understand the mechanisms regulating these processes. Despite the enormous strides in gene editing technology over the past decade using programmable nucleases, challenges remain in the field such as increasing the size of the targeting vector or minimizing off target genetic modifications. Understanding the properties of cis and trans factors affecting the efficiency of gene targeting through classical HR can yield mechanistic insights that have positive implications for other gene targeting technologies.

While to goal of this manuscript is to advance the mechanistic understanding of how properties of the Myh9 locus enable such high gene targeting efficiencies through HR, the low experimental robustness and interpretation presented in this manuscript limit the validity of the conclusions drawn. It is by these qualities that this manuscript as submitted is not appropriate for publication in PLOS One, and descriptions of these qualities are outlined below.

First, it is unclear in the written text what the authors aim to convey from the sequence comparisons across different species. Is there prior evidence that sequence conservation predicts strong recombination effects? Is there evidence that HR is high or low in ESCs of other species? If, for instance, HR efficiency is comparable in both rat and mouse ESCs (PMID: 20703227, PMID: 21151976), can conserved sequences be found across both loci? If there are discrepancies in HR efficiency across two species despite conserved sequences within syntenic elements, this could suggest differential epigenetic effectors. The authors can incorporate efficiency data from other ESC studies and better frame the interpretations drawn from their analysis. Furthermore, the data presented in Table 1, specifically the percent sequence similarity among the homology arms should be analyzed separately for the 5’ and 3’ arms. Figure 1A, or any of the other gene diagrams are not to scale and should indicate this.

Next, the authors should report on how many different ESC, iPSC, and MEF clonal lines were used to produce the data presented in Figure 2, as well as what passage number. If a single clonal cell line were used for each, this limits the generalizability of their findings. Furthermore, if these properties change over prolonged passaging of cell lines, this could also explain effects on efficiency. An ideal experiment to truly compare epigenetic states would be to conduct HR experiments on cell types derived from the same genetic individual. For instance, the authors could in vitro differentiate an ESC line into somatic cells such as fibroblasts, then reprogram them into iPSCs, and perform HR experiments on all three epigenetic states. Methodologically, the PCR should be designed in a way to detect a PCR product produced from the wild type alleles, as opposed to seeing no PCR product amplified. When no PCR band is detected for that sample, it could have resulted from a failed PCR run. The agarose gel images should be presented. Additionally, the authors should indicate where and whether the qPCR primers and antibody epitope account for exon 2.

Lastly, what is the contribution of cell cycle of each cell type to the HR efficiency? Could a difference in cell cycling rate between the ESC and iPSC lines examined explain the difference in the rate of successful targeting?

In sum, this manuscript falls short of the technical standards and logical conclusions expected from PLOS publications. Until the authors address the shortcomings described above and write a coherent story based on tenable data, this work should not be published as presented.

Reviewer #2: Tan and co-workers explore the efficiency of traditional gene targeting of the murine Myh9 locus. The work in the manuscript is focused on the mouse Myh9 locus, however, the discussion includes speculation about other species with relatively shallow bioinformatics analysis. Conversely, detailed analysis of the murine Myh9 locus and the sequence of the homology arms lacks any sizable detail. While the identification of the murine Myh9 locus as a mitotic recombination hotspot would be of interest for the field of genome editing, especially for those trying to optimize gene targeting, the manuscript as-is lacks any in depth analysis. The manuscript is both highly speculative about the underlying cause of the high efficiency targeting, while being relatively shallow about the details of the actual molecular work that was done. The manuscript would be greatly improved if it was focused on the mouse Myh9 locus and provided greater details about the various targeting vectors utilized and the genome editing efficiencies observed.

Major Comments

1) The comparison of the Myh9 locus in various species does not provide any insight into the high targeting efficiency of the murine Myh9 locus. In this manuscript, only the mouse genome is targeted, and it is inappropriate to infer anything about the targeting potential of the Myh9 locus in these other species, and the contribution of the flanking genomic sequence and the species-specific Myh9 locus. Unless IPSCs from these various species are also used for homologous gene targeting of the Myh9 locus, there is little information to be gained from comparing the Myh9 loci.

2) Although many aspects of homologous directed targeting remain unknown, it is accepted that critical for homologous targeting is mitosis and proliferation. Therefore, the inability to target the Myh9 locus in MEFs, which have a lower proliferative index and readily senescence in culture, is not surprising. The lack of senescence and high proliferation rate of ESCs and IPSCs is critical for efficient homologous gene targeting. Therefore, the expression level of Myh9 in MEFs is inconsequential in this reviewer’s opinion. The proliferation rate/index of the IPSCs and ESCs used in this study should be compared.

3) With gene targeting efficiency over 50%, it would be expected that some of the clones to be homozygous for the targeted allele. For a locus specific report about homologous targeting, zygosity of targeting should be added to the manuscript.

4) The data presented in Table 4 has a good linear relationship between total homology length and targeting efficiency. As presented in the table, it is difficult to see that relationship.

5) The varying targeting efficiency between the three regions of the murine Myh9 locus discussed in Table 3 warrant some additional sequence analysis, such as GC content, repetitive elements, homology to other location in the murine genome, etc.

6) Was chromatin accessibility assessed for the Myh9 locus? Even if not in the actual ESC line used, it would be interesting to know the status of the chromatin/nucleosome in mESCs of the readily targetable Myh9 locus.

Minor Comments

1) Minor grammatical errors such as line 64: “…is rarely”

2) Line 142/143; the purpose of the sub-cloning is unclear.

3) The choice of using the 129-derived BAC DNA as a template for the homology arms and the use of B6129F1 ES cells for electroporation is unclear given that isogenic homology arms are believed to increase targeting efficiency (Line 68).

4) If available, the targeting efficiency of the Myh9 locus in the absence of negative selection (ganciclovir) would be a useful comparison, since negative selection in practice rarely results in the enrichment of correctly targeted clones that are conceptually expected.

5) Source/method of IPSCs generation not indicated.

Reviewer #3: In this study, the authors intend to investigate the molecular biology underlying for the pronounced high gene targeting frequency at Myh9 gene locus. Numerous factors which affect the HR efficiency were investigated, including chromosome position, transcriptional activity, targeted loci, and the length of the homology arm. And details see the attachment.

6. PLOS authors have the option to publish the peer review history of their article (what does this mean?). If published, this will include your full peer review and any attached files.

Reviewer #1: No

Reviewer #2: No

Reviewer #3: No

---

## [Author Response · Author response to Decision Letter 0]

20 Nov 2019

Response to Reviewers

Reviewer #1: In the manuscript entitled “Investigation of the molecular biology underlying the pronounced high gene targeting frequency at Myh9 gene locus in mouse embryonic stem cells”, the authors perform homologous recombination (HR) in mouse embryonic stem cells (ESCs), induced pluripotent stem cells (iPSCs), and embryonic fibroblasts (MEFs) to investigate the molecular features of the Myh9 gene locus that are responsible for the notably high efficiency of gene targeting. They perform comparative genetic sequence analysis of the Myh9 gene locus to illustrate a low degree of sequence conservation, which in turn suggests there are no unique genetic properties at these loci that affect recombination efficiency. By targeting their vector into the exon 2 region of Myh9 in mouse ESCs, iPSCs, and MEFs, the authors demonstrate a variable HR efficiency that is cell type-dependent but transcriptionally and translationally independent. The efficiency of HR also declines as the targeting vector is shifted downstream of exon 2, highlighting the contributions of local regions to recombination efficiency. Finally, the authors investigate the effects of modifying the lengths of the homology arms and discover an optimal ratio of 5’ to 3’ arm lengths for effective HR. Because gene editing has proven itself as an indispensable tool in biomedical research, primarily in enabling the generation of genetic models of development and disease, it is a worthwhile effort to understand the mechanisms regulating these processes. Despite the enormous strides in gene editing technology over the past decade using programmable nucleases, challenges remain in the field such as increasing the size of the targeting vector or minimizing off target genetic modifications. Understanding the properties of cis and trans factors affecting the efficiency of gene targeting through classical HR can yield mechanistic insights that have positive implications for other gene targeting technologies. While to goal of this manuscript is to advance the mechanistic understanding of how properties of the Myh9 locus enable such high gene targeting efficiencies through HR, the low experimental robustness and interpretation presented in this manuscript limit the validity of the conclusions drawn. It is by these qualities that this manuscript as submitted is not appropriate for publication in PLOS One, and descriptions of these qualities are outlined below.

Answer: All of the authors thank the reviewer for the comments on this manuscript. In the revised version, we perform more detailed bioinformatic analysis, remove the not well addressed part conducted in mESCs, miPSCs and MEFs, and also provide necessary results. Moreover, we answer the concerns/questions raised by the reviewer in a point-to-point way as following.

Question #1: A: First, it is unclear in the written text what the authors aim to convey from the sequence comparisons across different species. B: Is there prior evidence that sequence conservation predicts strong recombination effects? C: Is there evidence that HR is high or low in ESCs of other species? If, for instance, HR efficiency is comparable in both rat and mouse ESCs (PMID: 20703227, PMID: 21151976), can conserved sequences be found across both loci? D: If there are discrepancies in HR efficiency across two species despite conserved sequences within syntenic elements, this could suggest differential epigenetic effectors. The authors can incorporate efficiency data from other ESC studies and better frame the interpretations drawn from their analysis. E: Furthermore, the data presented in Table 1, specifically the percent sequence similarity among the homology arms should be analyzed separately for the 5’ and 3’ arms. Figure 1A, or any of the other gene diagrams are not to scale and should indicate this.

Answer #1: We thank the reviewer for the comments. The questions in this paragraph are split into multiple ones and answered separately.

A: To our knowledge, this is the first time that such a high homologous recombination efficiency was reported and extensively investigated in mammalian cells based on traditional homologous recombination-mediated gene targeting [Liu T et al, PLoS One. 2018]. This property has facilitated the generation of a series of mouse models [Wang A et al, Proc Natl Acad Sci U S A. 2010; Zhang Y et al, Blood. 2012]. This efficiency is even much higher than using much longer homologous arms, e.g bacterial artificial chromosomes (BAC) DNA (6%, Testa G et al. Nat Biotechnol. 2003, 21(4):443-7), or designer nucleases-facilitated (ZFN, TALEN, and CRISPR) gene-targeting (≥20%, Casola S. Methods Mol Biol. 2010, 667:145-63). We have consulted a number of experts who study homologous recombination, but have not gained any useful clues on investigating the underlying mechanisms for this unique phenomenon. HR efficiency is collectively decided by many factors such as the target locus itself, isogenic DNA, length of homologous arms, transcription activity of the target locus, and the status of the ES cells [Ledermann B, Exp Physiol. 2000, 85(6):603-13]. Additionally, previous study indicated that meiotic recombination hotspots in mice are clustered with the major histocompatibility complex (MHC) region (Paul P et al. DNA Repair. 2016, 40:47-56), implying the genetic context matters. Meanwhile, it was suggested that comparative analysis of DNA sequences from multiple species at varying evolutionary distances is a powerful approach for identifying sequences that are unique for a given organism [Frazer KA, et al. Genome Res. 2003;13(1):1-12]. As an initial effort, we compared the genes flanking the Myh9 locus and the Myh9 sequences of mouse used to generate the homology arms with those of other species. The aim of these comparisons is to examine whether the Myh9 gene locus has undescribed and unique properties among different species including the conservation in terms of genetic context and whether there are distinct cis-elements, resembling LoxP, FRT, and other sequences, contributing to this high HR efficiency. The locus and sequence comparison could at least rule out this possibility or provide clues. 

B: There is no evidence indicating that sequence conservation predicts higher recombination rate, but there are indications that sequence divergence has an inhibitory effect on homologous recombination [Lukacsovich T et al. Genetics. 1999, 57(1):43-56; Opperman R et al. Genetics. 2004, 168(4):2207-15]. 

C: Mouse is the only species, in which ES cells have been extensively used in gene-targeting experiments. Germline-competent ES cells have also been generated in rats, but very few gene-targeting experiments have been reported. ES cell lines from some other mammalian species have been reported, but they have not been proven to be able to transmit through the germline, and therefore not used much in gene knockout experiment. No reports or data are available for comparing the targeting efficiency in mouse and rat embryonic stem cells. In general, the ES cells-based gene targeting efficiency is around 1-10% in the species of mouse [Brown AC et al. Cytotechnology. 2006, 51(2):81-8]. As far as the cases mentioned by the reviewer, it seems the HR efficiencies in those two cell types are comparable without considering the strategies and conditions used [Doetschman T et al. Proc Natl Acad Sci U S A. 1988, 85(22):8583-7; Donehower LA et al. Nature. 1992, 356(6366):215-21; Meek S et al. PLoS One. 20105(12):e14225; Tong C et al. Nature. 2010, 467(7312):211-3], in which a targeting efficiency of 1-3.7% can be achieved in the ES cells from both species. 

D: It is worth mentioning that most targeting efficiencies reported in the literature are within an expected range, together with multiple determinants for the efficiency, make it difficult to compare these cases. Likewise, a tight and direct relationship between epigenetic factors and homologous recombination(HR)-based gene targeting remains lacking, although there are a lot of studies indicating the contribution of these epigenetic factors to HR-directed DNA repair [Vélez-Cruz R et al. Genes Dev. 2016, 30(22):2500-2512; Lang F et al. Proc Natl Acad Sci U S A. 2017, 114(41):10912-10917; Chakraborty U et al. Genetics. 2019, 212(4):1147-1162]. Undoubtedly, the property of high gene targeting frequency at the MyH9 gene locus reported here facilitates to investigate those interesting questions in the future. 

E: Additionally, the bioinformatic analysis of the similarity among the homology arms is further conducted and presented, and the claim of the gene diagrams not to scale is added in the revised version.

Question #2: A: Next, the authors should report on how many different ESC, iPSC, and MEF clonal lines were used to produce the data presented in Figure 2, as well as what passage number. If a single clonal cell line were used for each, this limits the generalizability of their findings. Furthermore, if these properties change over prolonged passaging of cell lines, this could also explain effects on efficiency. B: An ideal experiment to truly compare epigenetic states would be to conduct HR experiments on cell types derived from the same genetic individual. For instance, the authors could in vitro differentiate an ESC line into somatic cells such as fibroblasts, then reprogram them into iPSCs, and perform HR experiments on all three epigenetic states. C: Methodologically, the PCR should be designed in a way to detect a PCR product produced from the wild type alleles, as opposed to seeing no PCR product amplified. When no PCR band is detected for that sample, it could have resulted from a failed PCR run. The agarose gel images should be presented. Additionally, the authors should indicate where and whether the qPCR primers and antibody epitope account for exon2.

Answer #2: We thank the reviewer for the comment. A: Since the similar concern was raised by all reviewers, we remove the part involving the experiment conducted in miPSCs and MEFS after communicating with the Academic editor, Dr Hu. All mouse ESC gene-targeting experiments were conducted using passage 14 (P14) V6.5 ES cell line. Our transgenic core has frozen hundreds of vials of P14 V6.5 ES cells, and we thaw a new vial for each construct. Therefore, all constructs were done using the same passage cells and the cell cycle/proliferation rate should be very similar. B: miPSC and MEF data have been removed. C: The long-range PCR using one primer located at the Neomycin selection maker and one primer resided in the outside of the short arm, has widely been used to identify the gene targeting recombinants in our previous and other studies [Liu T et al, PLoS One. 2018, 13(2):e0192641; Zhou D et al. J Vis Exp. 2018, (141); Lay JM et al. Transgenic Res. 1998, 7(2):135-40; Anastassiadis K et al. Methods Enzymol. 2013533:133-55; Fisher CL et al. Nucleic Acids Res. 2017, 45(21):e174; Sommer D et al. Nat Commun. 2014, 5:3045]. Furthermore, a positive and a negative control were included in each run of PCR to rule out the PCR failure. The PCR products were also sequenced to further confirm the targeting events. Additionally, representative agarose gel images are presented in the revised version, all the PCR primers have been listed in the supplementary table 1.

Question #3: A: Lastly, what is the contribution of cell cycle of each cell type to the HR efficiency? Could a difference in cell cycling rate between the ESC and iPSC lines examined explain the difference in the rate of successful targeting? B: In sum, this manuscript falls short of the technical standards and logical conclusions expected from PLOS publications. Until the authors address the shortcomings described above and write a coherent story based on tenable data, this work should not be published as presented.

Answer #3: We thank the reviewer for the comment. A: As stated above, we have removed the experiments conducted in miPSCs and MEFs per the permission of the academic editor, Dr. Hu, though our observation of the gene targeting efficiency in mouse ES cells higher than that in iPS cells is to some degree consistent with a previous study [Zou J et al. Cell Stem Cell. 2009, 5(1):97-110]. B: We believe that our revised version has been improved and should reach the standard of the journal and the response to the comments facilitates to understand the story of this manuscript.

Reviewer #2: Tan and co-workers explore the efficiency of traditional gene targeting of the murine Myh9 locus. The work in the manuscript is focused on the mouse Myh9 locus, however, the discussion includes speculation about other species with relatively shallow bioinformatics analysis. Conversely, detailed analysis of the murine Myh9 locus and the sequence of the homology arms lacks any sizable detail. While the identification of the murine Myh9 locus as a mitotic recombination hotspot would be of interest for the field of genome editing, especially for those trying to optimize gene targeting, the manuscript as is lacks any in depth analysis. The manuscript is both highly speculative about the underlying cause of the high efficiency targeting, while being relatively shallow about the details of the actual molecular work that was done. The manuscript would be greatly improved if it was focused on the mouse Myh9 locus and provided greater details about the various targeting vectors utilized and the genome editing efficiencies observed.

Answer: All of the authors thank the reviewer for the comments on this manuscript. In the revised version, we perform more detailed bioinformatic analysis, remove the not well addressed part conducted in mESCs, miPSCs and MEFs, and also provide necessary results. Moreover, we answer the concerns/questions raised by the reviewer in a point-to-point way as following.

Major Comments

Question #1): The comparison of the Myh9 locus in various species does not provide any insight into the high targeting efficiency of the murine Myh9 locus. In this manuscript, only the mouse genome is targeted, and it is inappropriate to infer anything about the targeting potential of the Myh9 locus in these other species, and the contribution of the flanking genomic sequence and the species-specific Myh9 locus. Unless IPSCs from these various species are also used for homologous gene targeting of the Myh9 locus, there is little information to be gained from comparing the Myh9 loci.

Answer #1: The authors thank the reviewer for this suggestion. 1) it was suggested that comparative analysis of DNA sequences from multiple species at varying evolutionary distances is a powerful approach for identifying sequences that are unique for a given organism [Frazer KA, et al. Genome Res. 2003;13(1):1-12]. 2) The aim of these comparisons is to examine whether the Myh9 gene locus has undescribed and unique properties among different species including the conservation in terms of genetic context and whether there exist unique elements like LoxP element contributing to this high HR efficiency. The locus and sequence comparison could at least rule out this possibility or provide other clues. 3) To our knowledge, it is the first time to report a so high homologous recombination efficiency at the Myh9 gene locus in mouse embryonic stem cells based on traditional gene targeting [Liu T et al, PLoS One. 2018], this property has facilitated the generation of a series of mouse models [Wang A et al, Proc Natl Acad Sci U S A. 2010; Zhang Y et al, Blood. 2012]. This efficiency is also much higher than that of even using longer homology arms, e.g bacterial artificial chromosomes (BAC) DNA (6%, Testa G et al. Nat Biotechnol. 2003, 21(4):443-7) and that of widely used safe harbor site ROSA26 in mouse embryonic stem cells (≥20%, Casola S. Methods Mol Biol. 2010, 667:145-63). We have even consulted with several experts in the field. Frankly to say, we could not get useful clues to investigate the underlying mechanisms for this unique phenomenon. HR efficiency is collectively decided by many factors such as the target locus itself, isogenic DNA, length of homologous arms, transcription activity of the target locus, and the status of the ES cells [Ledermann B, Exp Physiol. 2000, 85(6):603-13]. Additionally, previous study indicated that meiotic recombination hotspots in mice are clustered with the major histocompatibility complex (MHC) region (Paul P et al. DNA Repair. 2016, 40:47-56), implying the genetic context matters. As the initial effort, we compared the genes flanking the Myh9 locus and the mouse Myh9 sequences used to generate the homology arms with those of other species. 4) The detailed comparisons revealed that the mouse Myh9 gene locus is not unique in terms of genetic context including its flanking genes and gene length, while the sequences used to create the homology arms have higher similarity with that of Rat than with those of other species, moreover, the homology of the right arm sequence is higher than that of the left one. Additionally, the right arm is GC rich relative to the left arm and those from other species. Further investigations are warranted to determine whether these properties are associated with the GT efficiency.

Question #2): Although many aspects of homologous directed targeting remain unknown, it is accepted that critical for homologous targeting is mitosis and proliferation. Therefore, the inability to target the Myh9 locus in MEFs, which have a lower proliferative index and readily senescence in culture, is not surprising. The lack of senescence and high proliferation rate of ESCs and IPSCs is critical for efficient homologous gene targeting. Therefore, the expression level of Myh9 in MEFs is inconsequential in this reviewer’s opinion. The proliferation rate/index of the IPSCs and ESCs used in this study should be compared.

Answer #2: The authors thank the reviewer for this concern. Indeed, the cell cycle is an important factor affecting GT efficiency [Majumdar A et al. J Biol Chem. 2003, 278(13):11072-7]. Since the point could not well addressed in the study, we remove the part involving the experiment conducted in miPSCs and MEFS after communicating with the Academic editor, Dr Hu. 

Question #3): With gene targeting efficiency over 50%, it would be expected that some of the clones to be homozygous for the targeted allele. For a locus specific report about homologous targeting, zygosity of targeting should be added to the manuscript.

Answer #3: The V6.5 ESCs were derived from F1 hybrid mice, with 50% 129 and 50% C57BL/6, while the targeting constructs were made using 129 genomic DNA. Therefore, the constructs are isogenic to one of the two chromosomes. Importantly, targeting only one chromosome is advantageous in this case because homozygous knockout of Myh9 can cause abnormal proliferation and ESC morphology [Conti MA et al. J Biol Chem. 2004;279(40):41263-6], which can result in bias for colony picking. No ES clones with double knockout alleles were identified whether using PCR or Southern Blot methods [Wang A et al. Proc Natl Acad Sci U S A. 2010;107(33):14645-50; Zhang Y et al. Blood. 2012;119(1):238-50; Liu T et al. PLoS One. 2018;13(2):e0192641]. This information has been added in the revised version.

Question #4): The data presented in Table 4 has a good linear relationship between total homology length and targeting efficiency. As presented in the table, it is difficult to see that relationship.

Answer #4: Thanks for the useful suggestion. To better indicate the relationship between total length of homology arms and GT efficiency, we have reorganized the data and provided more detailed analysis. Indeed, there exists a linear relationship between those two, as shown in Figure 3 in the revised version. 

Question #5): The varying targeting efficiency between the three regions of the murine Myh9 locus discussed in Table 3 warrant some additional sequence analysis, such as GC content, repetitive elements, homology to other location in the murine genome, etc.

Answer #5: Thank the reviewer for this good suggestion. We performed further sequence analysis of the constructs targeting to different sites of the Myh9 gene locus, including GC content, CpG island, simple sequence repeat (SSR), SINE, LINE, or other DNA elements, as summarized in the table 3 in the revised version. Interestingly, there is a consistency between the percentage of SSR and SINE and the efficiency of GT. No significant similarity between the sequences used to create these targeting constructs and the sequences of other regions in the mouse genome was observed.

Question #6): Was chromatin accessibility assessed for the Myh9 locus? Even if not in the actual ESC line used, it would be interesting to know the status of the chromatin/nucleosome in mESCs of the readily targetable Myh9 locus.

Answer #6: Thank the reviewer for the comment. Frankly to say, the information about the status of the chromatin accessibility at the Myh9 locus in mESCs could not facilitate to explain the readily targetable Myh9 locus. On one hand, it has been suggested that the chromatin state influences the targeting or editing efficiency mediated by Triplex-forming oligonucleotides (TFOs) or genome editing tools [Macris MA et al. J Biol Chem. 2003 Jan 31;278(5):3357-62; van Rensburg R et al. Gene Ther. 2013 Feb;20(2):201-14; Daboussi F et al. Nucleic Acids Res. 2012 Jul;40(13):6367-79], while no evidence indicate the effect of chromatin accessibility on traditional HR-based gene targeting. On the other hand, earlier studies demonstrated that GT frequency is independent of target gene transcription [Johnson RS et al. Science. 1989,245(4923):1234-6; Yáñez RJ et al. Gene Ther. 1998, 5(2):149-59]. More importantly, our previous studies indicated that the expression level of the Myh9 gene in mESCs is considerable [Conti MA et al. J Biol Chem. 2004,279(40):41263-6; Wang A et al. Proc Natl Acad Sci U S A. 2010;107(33):14645-50; Zhang Y et al. Blood. 2012;119(1):238-50; Liu T et al. PLoS One. 2018;13(2):e0192641]. 

Minor Comments

1) Minor grammatical errors such as line 64: “…is rarely”

A: This error has been corrected. Furthermore, we have also corrected some other grammatical errors during revision. 

2) Line 142/143; the purpose of the sub-cloning is unclear.

A: The purpose of cloning PCR products into T-easy for sequencing is to make sure the GT recombinants identified by PCR are absolutely right ones. Because one primer is located at the 3’end of Neomycin and another one resides outside of the short arm, the sequencing result can confirm this.

3) The choice of using the 129-derived BAC DNA as a template for the homology arms and the use of B6129F1 ES cells for electroporation is unclear given that isogenic homology arms are believed to increase targeting efficiency (Line 68).

A: The V6.5 ESCs were derived from F1 hybrid mice, with 50% 129 and 50% C57BL/6, while the targeting constructs were made using 129 genomic DNA. Therefore, the constructs are isogenic to one of the two chromosomes. Importantly, targeting only one chromosome is advantageous in this case because homozygous knockout of Myh9 can cause abnormal proliferation and ESC morphology [Conti MA et al. J Biol Chem. 2004;279(40):41263-6], which can result in bias for colony picking. 

4) If available, the targeting efficiency of the Myh9 locus in the absence of negative selection (ganciclovir) would be a useful comparison, since negative selection in practice rarely results in the enrichment of correctly targeted clones that are conceptually expected.

A: We had no data for GT efficiency at the Myh9 gene locus in the absence of ganciclovir.

5) Source/method of IPSCs generation not indicated.

A: After communicating with the academic editor Dr. Hu, we decided to remove the part involving the experiments conducted in miPSCs and MEFs, thus, the source/method of iPSCs generation is no longer needed.

Reviewer #3: In this study, the authors intend to investigate the molecular biology underlying for the pronounced high gene targeting frequency at Myh9 gene locus. Numerous factors which affect the HR efficiency were investigated, including chromosome position, transcriptional activity, targeted loci, and the length of the homology arm. Some issues should be addressed or discussed in detail below.

A: The authors thank the reviewer for providing comments on this manuscript. The concerns or issues are addressed below and in the revised version.

Major comments:

1. Authors constructed 5’ and 3’ homologous arm from the 129/Sv genomic BAC clone DNA, and then the mouse cells from hybrid C57BL/6 and 129S6/SvEv were targeted. For sequence divergence among laboratory mice and the divergence imposed a major handicap in mutating the gene through homologous recombination (see HEIN TE RIELE, 1992, ref#15), it is possible that the cells line background affects the HR efficiency. The authors should sequence the target locus of cell lines to ensure that is consistent with the homologous arm sequences.

Answer #1: Thanks for this comment. The V6.5 ESCs were derived from F1 hybrid mice, with 50% 129 and 50% C57BL/6, while the targeting constructs were made using 129 genomic DNA. Therefore, the constructs are isogenic to one of the two chromosomes. Importantly, targeting only one chromosome is advantageous at least in this case because homozygous knockout of Myh9 can cause abnormal proliferation and ESC morphology [Conti MA et al. J Biol Chem. 2004;279(40):41263-6], which can result in bias for colony picking. Additionally, no ES clones with double alleles knocked out were identified whether using PCR or Southern Blot method.

2. Authors declared that “No animal experiments were conducted, no ethic permits were therefore needed for this report”, however, “MEFs were isolated and cultured embryonic day 9.5 to 10.5 (line 130, ref#29)”.

Answer #2: Thanks for reminding this point. We are sorry for not paying attention to it. After communicating with the academic editor Dr Hu, we decide to remove the part involving the experiments conducted miPSCs and MEFs, thus, the source/method of MEFs generation should be removed too. Thus, no animal experiments are indeed involved.

3. The authors did analyze the locus of the Myh9 gene in multiple species. However, there is no data showing the GT frequency in these species. Therefore, it’s hardly to reach the conclusion that low correlation with the high GT frequency at the Myh9 gene in term of chromosome position from the variable or conserved the locus Myh9 and its flanking genes. And several other types of genomic elements should be analyzed which may affect GT efficiency in Myh9, including CpG islands, simple repeats, microsatellites, DNA transposons, SINEs, LINEs et al.

Answer #3: Thanks for this good suggestion. According to the suggestion, we have performed further bioinformatic analysis on the sequences used to create construct targeting to the Myh9 gene exon2 in the mouse genome and the corresponding sequences from other species, the results are summarized and presented in the revised version.

4. The authors declared that the length and ratio of homologous arms affected the GT efficiency (Table 4). However, when targeting the Myh9 gene Exon 2, Intron 2 and Exon 3, why choose similar length of the homologous arms rather than the same length? And Dose the difference length and ratio of homologous arms also lead to the results?

Answer #4: Thanks for this inquiry. Firstly, a 11.5kb region spanning from the upstream of exon2 to the downstream of the exon3 was used to create those three targeting constructs. These constructs were made at different times. Due to the availability of restriction sites and PCR primer sequences, sometimes it is hard to make constructs with exact the same length. The GT efficiency is determined by multiple factors, such as the locus, the length of the homologous arms [Ledermann B, Exp Physiol. 2000, 85(6):603-13]. Thirdly, according to previous studies and our observation, the length and ratio of the homologous arms influences the GT efficiency. In current case, the length and ratio of the homologous arms of the constructs targeting to the Myh9 exon2, intron2 and exon3 was slightly increased, which were done unintentionally. Surprisingly, the GT efficiency is contrary to the length and ratio of the homologous arms (Figure 2 and table 2), suggesting other factors are responsible for this. 

Minor comments:

5. In Figure 1A, The Unch is short for? In Figure 1B shows the Divergence but the Figure legend is similarity.

Answer #5: Thanks for the inquiry and correction. The Unch is short for uncharacterized, as indicated in the figure legend. In Figure1B, both the percentage of divergence and similarity are indicated.

6. In Figure 2A, what does the black triangle indicates? In Figure 2B and 2D, the mRNA level and the protein level of NMHC IIA in ES cells were set as 100%, the SD should not exist in ES cells columns.

Answer #6: Thanks for the inquiry. The part involving the experiments conducted in mESCs, miPSCs and MEFs has been removed from the manuscript after communicating with the academic editor Dr Hu.

7. Authors should provide the data of identify recombination events

Answer #7: Thanks for the suggestion. Representative images of PCR identification of HR events were provided in the revised version.

8. The antibodies company should be labeled.

Answer #8: Thanks for the suggestion, but it is unnecessary since that part has been removed.

9. Many grammatical or spelling should be corrected thoroughly in the manuscript.

Answer #9: Thanks for this suggestion. The manuscript has been revised and the grammatical or spelling errors has been checked by a native speaker Dr Adelstein (NIH/NHLBI).

---

## [Decision Letter · Decision Letter 1]

10 Dec 2019

PONE-D-19-15918R1

Investigation of the molecular biology underlying the pronounced high gene targeting frequency at the Myh9 gene locus in mouse embryonic stem cells

PLOS ONE

Dear Dr Wang,

Thank you for submitting your manuscript to PLOS ONE. After careful consideration, we feel that it has merit but does not fully meet PLOS ONE’s publication criteria as it currently stands. Therefore, we invite you to submit a revised version of the manuscript that addresses the points raised during the review process.

We would appreciate receiving your revised manuscript by January 10, 2020. To enhance the reproducibility of your results, we recommend that if applicable you deposit your laboratory protocols in protocols.io, where a protocol can be assigned its own identifier (DOI) such that it can be cited independently in the future. For instructions see: http://journals.plos.org/plosone/s/submission-guidelines#loc-laboratory-protocols

We look forward to receiving your revised manuscript.

Kind regards,

Wenhui Hu, M.D., Ph.D.

Academic Editor

PLOS ONE

Reviewers' comments:

Reviewer's Responses to Questions

**Comments to the Author**

1. If the authors have adequately addressed your comments raised in a previous round of review and you feel that this manuscript is now acceptable for publication, you may indicate that here to bypass the “Comments to the Author” section, enter your conflict of interest statement in the “Confidential to Editor” section, and submit your "Accept" recommendation.

Reviewer #1: (No Response)

Reviewer #2: All comments have been addressed

Reviewer #3: (No Response)

2. Is the manuscript technically sound, and do the data support the conclusions?

Reviewer #1: Partly

Reviewer #2: Yes

Reviewer #3: Partly

3. Has the statistical analysis been performed appropriately and rigorously? 

Reviewer #1: Yes

Reviewer #2: Yes

Reviewer #3: Yes

4. Have the authors made all data underlying the findings in their manuscript fully available?

Reviewer #1: Yes

Reviewer #2: Yes

Reviewer #3: (No Response)

5. Is the manuscript presented in an intelligible fashion and written in standard English?

Reviewer #1: Yes

Reviewer #2: Yes

Reviewer #3: Yes

6. Review Comments to the Author

Reviewer #1: Tan et al. have removed and analyzed existing data to address reviewers’ concerns about the original draft of their manuscript. Their existing findings and conclusions relating gene targeting (GT) efficiency to cell type (i.e. ESC, iPSC, and MEF) have been removed to focus on the mechanisms underlying GT at the mouse Myh9 locus in ESCs. In response to reviewers’ suggestions, the authors have also analyzed genome sequence elements such as CG content, SSR, and SINE to consider functional elements possibly contributing to the high GT efficiency, specifically at the immediate regions flanking exon 2, which serve as the homology arms. They present this analysis on other species for comparison as well as present sequence elements for their other targeting sequences at intron 2 and exon 3. They have also presented gel images in the figures to show the outcome of their PCR results. Finally, the authors performed correlation analysis between total length of homology arms and targeting efficiency to quantitatively demonstrate a relationship.

The major concern remaining with the currently revised manuscript is the relevance of the genomics sequence comparison with other species. All three reviewers raised this concern and recognized that without GT efficiency data from other species, the comparison of genomic features is meaningless. The cross-species comparison of the locus sequence similarity is not an effective method to convey the mechanism of the high GT efficiency. One cannot draw any association between genetic features to function when data on function is completely not presented. This effort is better applied to compare genomics features among the Myh9 locus with other mouse loci that have regularly been used for GT in mouse ESCs, such as Rosa26, HPRT, or Col1a1, where the GT efficiency has been reported by other groups (PMID: 25143803, PMID: 17183668, PMID: 16400644, PMID: 19141541). While it would be more scientifically rigorous to for the authors to perform these GT experiments at these other loci themselves in the same cell lines they have reported on in this manuscript, they can perform meta-analysis by drawing associations based on data previously published by other labs based on their targeting constructs.

Despite this reviewer’s opinion that the species comparisons should not be performed and that the data do not support the authors’ conclusion, there are critical misinterpretations of the data as they are currently presented:

The methods section describes analyzing only the flanking 5’ and 3’ sequences of Myh9 exon 2 in all species. This is not presented in Figure 1A, which makes it rather misleading to think that Figures 1B, C, and D are based on Figure 1A. In Figure 1A, why is the orientation of the Myh9 loci reversed for the Pig and the Rabbit? This is confusing when the authors refer the right and left arm, rather than the 5’ and 3’ sequences. In Figure 1B, C, and D, the relationships are simply reflecting what is expected to be the relationship on a genome wide level, therefore one should not be surprised that mouse and rat bear such a high degree of similarity. Also, the algorithm for calculating percent identity and divergence should be clearly described for the reader.

Figure 2: PCR data is missing for intron 2.

Figure 3 is mislabeld as Figure 4.

The targeting construct series can benefit the reader by listing properties of each construct to the left of each diagram, such as combined length of homology arms, the ratio of 5’:3’, etc. the length of each arm should also be displayed next to each line.

Figure 3C is basically Figure 3D, but it is not appropriate to apply this type of graphing to this data set. The authors should only present scatter plots in the style of Figure 3D.

In Figure 3D, the authors perform a quantitative measure of the relationship between total length of homology arms and GT efficiency. They can test the significance of this correlation.

The authors should also perform similar quantifiable analysis using the ratio of 5’:3’ lengths, GC content, SSR, and SINE of total, and individual homology arms. These analyses may provide some more insight into the mechanisms underlying the respective GT efficiencies.

Table 4. Why is only one repeat shown? Or is this the sum of 2 repeats? Authors should specify.

Altogether, the manuscript as presented lack sufficient data to model a hypothesis, much less support a detailed, mechanistic understanding of why the Myh9 locus at exon 2 is such an efficient site for GT. The suggested, more in-depth analyses could bring this manuscript to a body of work suitable for publication in PLOS One.

Reviewer #2: The manuscript is greatly improved, and is much more focused on the underlying data and findings. The level of detail provided will allow others to potentially use the murine Myh9 locus as a site to further optimize gene targeting. My only remaining concern (which was shared by other initial reviewers) is the comparison of the murine Myh9 locus with that of other species. With zero gene targeting information for the Myh9 locus in these other species, this is just a distraction for the reader. The inbred mouse is an invaluable tool in biomedical research, and this manuscript should focus on the murine Myh9 locus without speculating about orthologous sequence and how that relates or does NOT relate to the gene targeting. Furthermore, it is misleading to assume that "No unique features and low sequence similarity are found in the Myh9 gene" when comparing to other Myh9 loci for which there is zero gene targeting information. Perhaps a better comparison would be to randomly sample other murine paralogs or similar length/structure loci to see if there is something unique to the murine Myh9 locus that affords such high gene targeting efficiency.

Reviewer #3: Tan et al have partially addressed some concerns issued on their manuscript, "Investigation of the molecular biology underlying the pronounced high gene targeting frequency at the Myh9 gene locus in mouse embryonic stem cells." The reasons for similar length of the homologous arms have been properly elucidated. And the further bioinformatic analysis have been performed. However, some issues still should be addressed or discussed to make the manuscript fit for publication in plos one.

Comment-1

As mentioned before, the authors analyzed the locus of the Myh9 gene in multiple species. While there is no data showing the GT frequency in these species. Therefore, it’s hardly to reach the conclusion that low correlation with the high GT frequency at the Myh9 gene in term of chromosome position from the variable or conserved the locus Myh9 and its flanking genes.

Comment-2

The authors used the V6.5 ESCs which were derived from F1 hybrid mice, with 50% 129S6/SvEv and 50% C57BL/6 for gene targeting, and declared the constructs are isogenic to one of the two chromosomes, while there is no data shows the homology between 129S6/SvEv and C57BL/6 in this loci. Is there evidence that HR is occurs only in the Myh9 loci of 129S6/SvEv? If not, maybe there were some homozygous clones for the targeted allele, which may result in bias for colony picking, as the authors mention, for homozygous knockout of Myh9 can cause abnormal proliferation and ESC morphology. If homology is low, is the targeting efficiency of the Myh9 loci in C57BL/6 as high as 129S6/SvEv?

Comment-3

Unformal writing of gene name such as line 40 “…Myh9 (Italic), line 170 “…MYH9” and Figure 1A. In Figure 3C, the units of vertical and horizontal axis should be provided. In Figure 3D, as targeting efficiency will not be lower than 0, it is more appropriate to set the minimum value of the vertical axis to 0.

7. PLOS authors have the option to publish the peer review history of their article (what does this mean?). If published, this will include your full peer review and any attached files.

Reviewer #1: No

Reviewer #2: No

Reviewer #3: No

---

## [Author Response · Author response to Decision Letter 1]

23 Jan 2020

Review Comments to the Author

Reviewer #1: Tan et al. have removed and analyzed existing data to address reviewers’ concerns about the original draft of their manuscript. Their existing findings and conclusions relating gene targeting (GT) efficiency to cell type (i.e. ESC, iPSC, and MEF) have been removed to focus on the mechanisms underlying GT at the mouse Myh9 locus in ESCs. In response to reviewers’ suggestions, the authors have also analyzed genome sequence elements such as CG content, SSR, and SINE to consider functional elements possibly contributing to the high GT efficiency, specifically at the immediate regions flanking exon 2, which serve as the homology arms. They present this analysis on other species for comparison as well as present sequence elements for their other targeting sequences at intron 2 and exon 3. They have also presented gel images in the figures to show the outcome of their PCR results. Finally, the authors performed correlation analysis between total length of homology arms and targeting efficiency to quantitatively demonstrate a relationship.

Answer: All the authors thank the reviewer for appreciating the improvement of the manuscript.

The major concern remaining with the currently revised manuscript is the relevance of the genomics sequence comparison with other species. All three reviewers raised this concern and recognized that without GT efficiency data from other species, the comparison of genomic features is meaningless. The cross-species comparison of the locus sequence similarity is not an effective method to convey the mechanism of the high GT efficiency. One cannot draw any association between genetic features to function when data on function is completely not presented. This effort is better applied to compare genomics features among the Myh9 locus with other mouse loci that have regularly been used for GT in mouse ESCs, such as Rosa26, HPRT, or Col1a1, where the GT efficiency has been reported by other groups (PMID: 25143803, PMID: 17183668, PMID: 16400644, PMID: 19141541). While it would be more scientifically rigorous to for the authors to perform these GT experiments at these other loci themselves in the same cell lines they have reported on in this manuscript, they can perform meta-analysis by drawing associations based on data previously published by other labs based on their targeting constructs.

Answer: Thanks to the reviewer for this good suggestion! After serious consideration, we decide to remove the comparison of the Myh9 gene locus and the sequences used to create the construct targeting the Myh9 locus with the counterparts from other species. On the one hand, it more seems to be negative data; a functional confirmation of these genetic feature is lacking, on the other hand. According to the suggestion from this reviewer, in the new version, we collected related information about gene targeting at the Rosa26, HRPT, ColA1 loci suggested by the reviewer. The summarization and comparison of gene targeting events and genetic sequences at these loci with those at the Myh9 locus. These contents were described and presented in the new version. Though no common genetic characteristics could be identified among those four loci, a typical 6kb total length for the homologous arms, a 2:1 ratio for the length of the 5’ to 3’ homologous arm, similar and 50% GC-content for the 5’ and 3’ arms, seemed to have better performance, as further confirmed in our subsequent experiments.

Despite this reviewer’s opinion that the species comparisons should not be performed and that the data do not support the authors’ conclusion, there are critical misinterpretations of the data as they are currently presented:

Answer: Thanks a lot. Since the comparison and related data were removed, there was no this issue in the revised version.

The methods section describes analyzing only the flanking 5’ and 3’ sequences of Myh9 exon 2 in all species. This is not presented in Figure 1A, which makes it rather misleading to think that Figures 1B, C, and D are based on Figure 1A. In Figure 1A, why is the orientation of the Myh9 loci reversed for the Pig and the Rabbit? This is confusing when the authors refer the right and left arm, rather than the 5’ and 3’ sequences. In Figure 1B, C, and D, the relationships are simply reflecting what is expected to be the relationship on a genome wide level, therefore one should not be surprised that mouse and rat bear such a high degree of similarity. Also, the algorithm for calculating percent identity and divergence should be clearly described for the reader.

Answer: Thanks to the reviewer for this concern. Similarly, the part related to the comparison and analysis of the Myh9 locus and sequences with those of other species has been removed, the concern from the reviewer was not an issue.

Figure 2: PCR data is missing for intron 2.

Answer: Because a representative image for PCR identification of targeting events at the Myh9 intron2 site has been presented in our previous report [Liu T et al. PLos One, 2018], similar result was therefore omitted in this study.

Figure 3 is mislabled as Figure 4.

Answer: Thanks for the correction. All the figures have been reorganized and marked clearly.

The targeting construct series can benefit the reader by listing properties of each construct to the left of each diagram, such as combined length of homology arms, the ratio of 5’:3’, etc. the length of each arm should also be displayed next to each line.

Answer: Thanks for the good suggestion. Related information has been added and this figure has been modified.

Figure 3C is basically Figure 3D, but it is not appropriate to apply this type of graphing to this data set. The authors should only present scatter plots in the style of Figure 3D.

Answer: Thanks for the good suggestion. This graph has been modified in the new version.

In Figure 3D, the authors perform a quantitative measure of the relationship between total length of homology arms and GT efficiency. They can test the significance of this correlation.

Answer: Thanks for the suggestion. The test of the correlation significance (Represented by p<0.01) between the total length of homologous arms and the GT efficiency has been performed and presented in the revised version.

The authors should also perform similar quantifiable analysis using the ratio of 5’:3’ lengths, GC content, SSR, and SINE of total, and individual homology arms. These analyses may provide some more insight into the mechanisms underlying the respective GT efficiencies.

Answer: Thanks a lot for this good suggestion. The genetic feature analysis of the sequences used for creating different length homologous arms has been conducted, which is summarized and presented in the new version.

Table 4. Why is only one repeat shown? Or is this the sum of 2 repeats? Authors should specify.

Answer: Thanks for raising this concern. This point has been specified in the figure legends of the new version.

Altogether, the manuscript as presented lack sufficient data to model a hypothesis, much less support a detailed, mechanistic understanding of why the Myh9 locus at exon 2 is such an efficient site for GT. The suggested, more in-depth analyses could bring this manuscript to a body of work suitable for publication in PLOS One.

Answer: Thanks a lot for the comments. According to the comments and suggestions from this reviewer, the contents have been adjusted, the related information has been provided, and the figures have been modified, therefore, the manuscript should be greatly improved and reach to the requirement for the journal of PLoS One.

Reviewer #2: The manuscript is greatly improved, and is much more focused on the underlying data and findings. The level of detail provided will allow others to potentially use the murine Myh9 locus as a site to further optimize gene targeting. My only remaining concern (which was shared by other initial reviewers) is the comparison of the murine Myh9 locus with that of other species. With zero gene targeting information for the Myh9 locus in these other species, this is just a distraction for the reader. The inbred mouse is an invaluable tool in biomedical research, and this manuscript should focus on the murine Myh9 locus without speculating about orthologous sequence and how that relates or does NOT relate to the gene targeting. Furthermore, it is misleading to assume that "No unique features and low sequence similarity are found in the Myh9 gene" when comparing to other Myh9 loci for which there is zero gene targeting information. Perhaps a better comparison would be to randomly sample other murine paralogs or similar length/structure loci to see if there is something unique to the murine Myh9 locus that affords such high gene targeting efficiency.

Answer: Thanks to the reviewer for appreciating the improvement of this manuscript and providing other useful suggestions. We have removed the comparison of the Myh9 locus/Sequences from multiple species. Instead of these, we collected, organized and compared the targeting efficiency and the genetic features of several loci often used for site-specific integration, as also suggested by other reviewers. The new content is provided in the revised version.

Reviewer #3: Tan et al have partially addressed some concerns issued on their manuscript, "Investigation of the molecular biology underlying the pronounced high gene targeting frequency at the Myh9 gene locus in mouse embryonic stem cells." The reasons for similar length of the homologous arms have been properly elucidated. And the further bioinformatic analysis have been performed. However, some issues still should be addressed or discussed to make the manuscript fit for publication in plos one.

Answer: Thanks to the reviewer for appreciating the improvement of this manuscript and providing other useful comments.

Comment-1

As mentioned before, the authors analyzed the locus of the Myh9 gene in multiple species. While there is no data showing the GT frequency in these species. Therefore, it’s hardly to reach the conclusion that low correlation with the high GT frequency at the Myh9 gene in term of chromosome position from the variable or conserved the locus Myh9 and its flanking genes.

Answer: Thanks for raising this concern. We have deleted the comparison of the Myh9 locus/Sequences from multiple species. Instead of these, we collected, organized and compared the targeting efficiency and the genetic features of several loci often used for site-specific integration, as also suggested by other reviewers. The new content is provided in the revised version.

Comment-2

The authors used the V6.5 ESCs which were derived from F1 hybrid mice, with 50% 129S6/SvEv and 50% C57BL/6 for gene targeting, and declared the constructs are isogenic to one of the two chromosomes, while there is no data shows the homology between 129/Sv and C57BL/6 in this loci. Is there evidence that HR is occurs only in the Myh9 loci of 1296/Sv? If not, maybe there were some homozygous clones for the targeted allele, which may result in bias for colony picking, as the authors mention, for homozygous knockout of Myh9 can cause abnormal proliferation and ESC morphology. If homology is low, is the targeting efficiency of the Myh9 loci in C57BL/6 as high as 129S/Sv?

Answer: Thanks for raising this concern. The BAC DNA used as the template for obtaining the homologous arms is derived from a 129/Sv genetic background. Importantly, the blast of the whole sequences used for creating the homologous arms indicated that the sequences are100% matching with these from a C57BL/6 genetic background (using the NCBI/Blast program). In other words, the sequences used for creating the targeting constructs are isogenic to both chromosomes. Likewise, no homozygous ES clones were identified by either PCR or Southern Blot.

Comment-3

Unformal writing of gene name such as line 40 “…Myh9 (Italic), line 170 “…MYH9” and Figure 1A. In Figure 3C, the units of vertical and horizontal axis should be provided. In Figure 3D, as targeting efficiency will not be lower than 0, it is more appropriate to set the minimum value of the vertical axis to 0.

Answer: Thanks for these corrections. We have corrected all of them.

---

## [Decision Letter · Decision Letter 2]

17 Feb 2020

PONE-D-19-15918R2

Investigation of the molecular biology underlying the pronounced high gene targeting frequency at the Myh9 gene locus in mouse embryonic stem cells

PLOS ONE

Dear Dr Wang,

Thank you for submitting your manuscript to PLOS ONE. After careful consideration, we feel that it has merit but does not fully meet PLOS ONE’s publication criteria as it currently stands. Therefore, we invite you to submit a revised version of the manuscript that addresses the minor points raised during the second review process.

We would appreciate receiving your revised manuscript by March 1, 2020. To enhance the reproducibility of your results, we recommend that if applicable you deposit your laboratory protocols in protocols.io, where a protocol can be assigned its own identifier (DOI) such that it can be cited independently in the future. For instructions see: http://journals.plos.org/plosone/s/submission-guidelines#loc-laboratory-protocols

We look forward to receiving your revised manuscript.

Kind regards,

Wenhui Hu, M.D., Ph.D.

Academic Editor

PLOS ONE

Reviewers' comments:

Reviewer's Responses to Questions

**Comments to the Author**

1. If the authors have adequately addressed your comments raised in a previous round of review and you feel that this manuscript is now acceptable for publication, you may indicate that here to bypass the “Comments to the Author” section, enter your conflict of interest statement in the “Confidential to Editor” section, and submit your "Accept" recommendation.

Reviewer #1: All comments have been addressed

Reviewer #2: All comments have been addressed

Reviewer #3: All comments have been addressed

2. Is the manuscript technically sound, and do the data support the conclusions?

Reviewer #1: Yes

Reviewer #2: Yes

Reviewer #3: Partly

3. Has the statistical analysis been performed appropriately and rigorously? 

Reviewer #1: Yes

Reviewer #2: Yes

Reviewer #3: Yes

4. Have the authors made all data underlying the findings in their manuscript fully available?

Reviewer #1: Yes

Reviewer #2: Yes

Reviewer #3: Yes

5. Is the manuscript presented in an intelligible fashion and written in standard English?

Reviewer #1: Yes

Reviewer #2: Yes

Reviewer #3: Yes

6. Review Comments to the Author

Reviewer #1: (No Response)

Reviewer #2: The manuscript in its second revised form is greatly improved. The removal of speculative gene targeting of the Myh9 locus in other species from the main manuscript places focus on the actual work done. Supplementary Figure 1 is poor resolution (and completely unnecessary). While I understand that the multi-species Myh9 locus alignment was time consuming to generate, it really does not add anything of value to the manuscript.

Figure 2C and 2D are the exact same data. there is no need to present the same data twice, just add a trend line to 2C (scale on 2D is nonsensical, -20% targeting efficiency?)

throughout the paper the Myh9 introns and exons are capitalized and written both with and without a space before the number (e.g. Exon2, Exon 2, Intron2, Intron2, etc). Intent is obvious, but it should be at least consistent within the manuscript. Decision to capitalize intron and exon is questionable, but not a scientific issue.

Reviewer #3: Tan et al. have removed the comparison of the Myh9 locus from multiple species, and according to reviewers’ suggestion, the authors compared the length of homologous arms and genomics elements (CG content, SSR, CpG islands, Line, SINE) of the Myh9 locus with other mouse loci (Rosa26, HPRT, and Col1a1) that have widely been used for GT to reveal characteristics potentially contributing to the high GT efficiency. They have also analyzed genomics features of the truncated homologous arms.

Comment-1

The author removed the comparison of the Myh9 locus from multiple species, however, retains the alignment of these sequences in Supplementary Figure 1, and it may more preferably to removed Supplementary Figure 1 together.

Comment-2

Without GT efficiency data from other mouse loci with different ration of homologous arms, it is inappropriate to infer 2:1 ratio for the length of the 5’ and 3’ homologous arms facilitate GT efficiency from the result of comparison the Myh9 locus with other loci.

Comment-3

As other reviewer’ suggestion, Figure 3C is basically Figure 3D, the authors should only present Figure 3D and removed Figure 3C. The concern remaining in the Figure 3D, as targeting efficiency will not be lower than 0, it is more appropriate to set the minimum value of the vertical axis to 0.

Comment-4

In Figure 2A, No.2 total length is 4.6 kb, and the right homologous arm of No.3 should be left aligned.

7. PLOS authors have the option to publish the peer review history of their article (what does this mean?). If published, this will include your full peer review and any attached files.

Reviewer #1: No

Reviewer #2: No

Reviewer #3: No

---

## [Author Response · Author response to Decision Letter 2]

21 Feb 2020

Response to reviewers

Reviewer #1: (No Response)

-Reviewer #2: The manuscript in its second revised form is greatly improved. The removal of speculative gene targeting of the Myh9 locus in other species from the main manuscript places focus on the actual work done. Supplementary Figure 1 is poor resolution (and completely unnecessary). While I understand that the multi-species Myh9 locus alignment was time consuming to generate, it really does not add anything of value to the manuscript.

Answer：All the authors thank the reviewer for appreciating the improvement of the revised manuscript. The supplementary Figure 1 has been removed from this manuscript.

-Figure 2C and 2D are the exact same data. there is no need to present the same data twice, just add a trend line to 2C (scale on 2D is nonsensical, -20% targeting efficiency?)

Answer: Thanks for your suggestion. Fig 2D has been deleted, and a trend line has been incorporated into Figure 2C.

-Throughout the paper the Myh9 introns and exons are capitalized and written both with and without a space before the number (e.g. Exon2, Exon 2, Intron2, Intron2, etc). Intent is obvious, but it should be at least consistent within the manuscript. Decision to capitalize intron and exon is questionable, but not a scientific issue.

Answer: Thanks for your correction. We have used exon2 or intron2 instead of other ones in the whole manuscript to make sure the consistency.

Reviewer #3: Tan et al. have removed the comparison of the Myh9 locus from multiple species, and according to reviewers’ suggestion, the authors compared the length of homologous arms and genomics elements (CG content, SSR, CpG islands, Line, SINE) of the Myh9 locus with other mouse loci (Rosa26, HPRT, and Col1a1) that have widely been used for GT to reveal characteristics potentially contributing to the high GT efficiency. They have also analyzed genomics features of the truncated homologous arms.

Answer: All the authors thank the reviewer for appreciating the improvement of the manuscript.

Comment-1 The author removed the comparison of the Myh9 locus from multiple species, however, retains the alignment of these sequences in Supplementary Figure 1, and it may more preferably to removed Supplementary Figure 1 together.

Answer: Thanks a lot for your suggestion, Supplementary Figure 1 has been removed.

Comment-2 Without GT efficiency data from other mouse loci with different ratio of homologous arms, it is inappropriate to infer 2:1 ratio for the length of the 5’ and 3’ homologous arms facilitate GT efficiency from the result of comparison the Myh9 locus with other loci.

Answer: Thanks a lot for your consideration, we have amended this speculation for other loci.

Comment-3 As other reviewer’ suggestion, Figure 3C is basically Figure 3D, the authors should only present Figure 3D and removed Figure 3C. The concern remaining in the Figure 3D, as targeting efficiency will not be lower than 0, it is more appropriate to set the minimum value of the vertical axis to 0.

Answer: Thanks a lot for your suggestion. Fig 2C and Fig2D has been incorporated into one, e.g Fig2C in the revised version, your concern has also been solved.

Comment-4 In Figure 2A, No.2 total length is 4.6 kb, and the right homologous arm of No.3 should be left aligned.

Answer: Thanks a lot for your correction, this error for construct No2 has been amended, while the information for construct No3 was right, because we deleted the exon2 and beyond (an about 0.5kb) to obtain this left truncated form of 3’arm. A note for this has been added in the legend for Figure2A.

---

## [Editor Report · Decision Letter 3]

24 Feb 2020

Investigation of the molecular biology underlying the pronounced high gene targeting frequency at the Myh9 gene locus in mouse embryonic stem cells

PONE-D-19-15918R3

Dear Dr. Wang,

We are pleased to inform you that your manuscript has been judged scientifically suitable for publication and will be formally accepted for publication once it complies with all outstanding technical requirements.

With kind regards,

Wenhui Hu, M.D., Ph.D.

Academic Editor

PLOS ONE
---

## [Editor Report · Acceptance letter]

11 Mar 2020

PONE-D-19-15918R3 

Investigation of the molecular biology underlying the pronounced high gene targeting frequency at the Myh9 gene locus in mouse embryonic stem cells 

Dear Dr. Wang:

I am pleased to inform you that your manuscript has been deemed suitable for publication in PLOS ONE. Congratulations! Your manuscript is now with our production department. 

With kind regards,

on behalf of

Dr. Wenhui Hu 

Academic Editor

PLOS ONE